# Potential Migration and Health Risks of Heavy Metals and Metalloids in Take-Out Food Containers in South Korea

**DOI:** 10.3390/ijerph21020139

**Published:** 2024-01-26

**Authors:** Yujin Han, Keunyoung Ryu, Nanju Song, Jinjong Seo, Insook Kang, Hyun-Jung Chung, Ran Park

**Affiliations:** 1Health and Environment Research Institute of Gwangju, 584, Mujin-daero, Seo-gu, Gwangju 61954, Republic of Korea; zetryu@korea.kr (K.R.); snj0137@korea.kr (N.S.); sjj21@hanmail.net (J.S.); hjin00@korea.kr (I.K.); pppp82@korea.kr (R.P.); 2Division of Food and Nutrition, Chonnam National University, Gwangju 61186, Republic of Korea; hchung@jnu.ac.kr

**Keywords:** food container, metal transfer, health risk, plastic containers, antimony migration

## Abstract

The consumption of take-out food has increased worldwide; consequently, people are increasingly being exposed to chemicals from food containers. However, research on the migration of metals from containers to food is limited, and therefore, information required to determine the health risks is lacking. Herein, the amount of transfer of nine metals and metalloids (Pb, Sb, Cd, Ge, Co, Mn, Sn, As, and Hg) from food containers to food in South Korea was assessed from take-out food containers classified into paper and plastic container groups. The sample containers were eluted over time by either warming with 4% acetic acid at 70 °C or cooling with 4% acetic acid at 100 °C /deionized water at 25 °C. It was analyzed using an inductively coupled plasma mass spectrometer and a direct mercury analyzer. The reliability of the quantitative results was verified by calculating the linearity, limit of detection, and limit of quantification. We found that the amount of metals and metalloids (Pb, Sb, Cd, and Co) eluting over time was highly significant in the plastic group. Regardless of the food simulant and elution time, the amount of Sb transferred from the food containers to food was substantially higher in the plastic (average concentration: 0.488–1.194 μg/L) than in the paper group (average concentration: 0.001–0.03 μg/L). Fortunately, all food containers were distributed at levels safe for human health (hazard index: 0.000–64.756%). However, caution is needed when warm food is added to food containers. Overall, our results provide baseline data for the management and use of take-out containers.

## 1. Introduction

Disposable food containers are frequently used in daily life for hygiene management and convenience. Particularly, they are widely used in cafés and restaurants to pack and deliver food. Ordering and delivering take-out food has increased rapidly with the development of the Internet and changes in household types, such as an increase in dual-income couples, single-person households, and elderly households. These changes have increased the use of disposable food containers. Additionally, the COVID-19 outbreak in 2020 has increased the significance of personal hygiene management and quarantine to prevent the spread of infectious diseases, consequently increasing the use of food delivery and disposable products [1]. Plastic or paper materials are commonly used as packaging materials for food [2]. Of the plastic materials, polypropylene (PP), polystyrene (PS), polyethylene terephthalate (PET), and polyethylene are commonly used materials in food containers [3].

Food packaging materials facilitate easy storage and the transport of food; however, there are concerns that harmful substances, such as heavy metals, may be transferred to food from these containers [4,5]. Heavy metals are generally ingested through inhalation into the lungs, possibly due to occupational factors or through oral modes. Other modes of entry are contaminated water and food, and food containers [6,7].

Laws have been designed in countries worldwide to regulate the exposure of heavy metals from packaging materials. The European Union manages food contact materials through the Food Contact Material Regulation (EC) No. 1935/2004 [8], and the US regulates food contact substances through a regulation listed in Title 21 of the Code of Federal Regulations and criteria that recognize safe status [9]. In China, food containers and packaging materials are managed according to the Food Safety Act [10], and in Japan, food container materials are managed according to the Food Sanitation Law and the Ministry of Health, Labor, and Welfare Law [11].

South Korea, the European Union, the US, China, and Japan are conducting material tests, and all countries, except Korea, are promoting a positive list of resin additives [12]. Additionally, stability tests are being performed on items containing packaged food. In Korea, migration tests are conducted to check and manage the extent to which Pb is transferred to food. In Europe, plastic–food contact materials are managed according to a specific migration limit, which is the maximum allowable amount of substances approved for use in foods and similar solvents. In the US, stability is confirmed through the cumulative estimated daily intake and acceptable daily intake, while China manages health safety based on specific migration limits. Japan estimates the daily intake and acceptable daily intake of substances from packaging materials [13]. Although the use of food containers is increasing worldwide, and various regulations are implemented to manage their use, research on substances that can be transferred from food containers is insufficient [14].

Some previous studies have revealed a wide range of heavy metals, organic contaminants, and microplastics in the hot leachate from disposable food containers [5,15,16,17,18,19,20]. However, little is known about the risks to human health from exposure to disposable paper and plastic products. In one study, large amounts of metals were found to be released from disposable food containers in hot water, and the excess lifetime cancer risk results for Ni and Be identified that the carcinogenic risk may be non-negligible with chronic exposure [18]. In a search for leaching metals from take-out food containers in China, the total non-carcinogenic risk was low, but the combined carcinogenic risk of Cd, Pb, Ni, and Co was estimated to be unacceptable under certain exposure frequencies [14].

There are relatively few studies on PET as a food container material [5,17,19,20]. Several studies present results on the effects of Sb release from PET bottle storage conditions. The eluted Sb content of Hungarian mineral water (non-carbonated; carbonated) packaged in PET material was found to be in the range from 210 to 290 mg/kg [19]. A study of U.S. bottled water detected between 0.095 and 0.521 ng/L of antimony and further confirmed that storage at higher temperatures released increased amounts of it [17]. In China, antimony and bisphenol A released from polyethylene terephthalate (PET) water bottles were identified [20]. After one week of storage, Sb was found to increase from 1.88 to 8.32 ng/L at 4 °C, 2.10 to 18.4 ng/L at 25 °C, and 20.3 to 2604 ng/L at 70 °C, while smaller amounts of BPA were reported at 0.26–18.7, 0.62–22.6, and 2.89–38.9 ng/L [20]. Their study looked at the leaching of metals from water in water bottles, which suggests the need for an approach using food simulants that are representative of common foods. Since most people are not aware of the specific materials used in food containers, there is a need to study the larger category of materials including plastic and paper. More importantly, the method of elution from the food container to the food should reflect the time and method of consumption of food from a typical food container, such as keeping it warm or storing it at room temperature and then consuming it.

In this study, we investigated the transfer amount of nine metals and metalloids (Pb, Sb, Cd, Ge, Co, Mn, Sn, As, and Hg) from food containers to food in South Korea. Particularly, we determined the differences in the amount of metal and metalloid migration over time from commonly used plastic and paper food containers. Overall, our study monitored the safety of disposable food containers produced and distributed in Korea and compared the hazardous substances in plastic and paper containers to determine which material would be better in materials used for foods for human consumption. Additionally, this study contributes to improving the use of appropriate food packaging containers by confirming that the content of metals and metalloids eluted by different methods varies depending on the elution time and material.

## 2. Materials and Methods

### 2.1. Sample Collection

This study was conducted at the Health and Environment Research Institute of Gwangju Metropolitan City, Republic of Korea. In total, 120 disposable containers distributed across South Korea were collected and classified into paper (*n* = 60) and plastic (*n* = 60) groups according to the packaging material. Containers with different manufacturing dates and manufacturers were collected.

### 2.2. Reference Standard and Reagents

Certified reference standards for inductively coupled plasma mass spectrometry (ICP–MS) were purchased from Accu Standard (New Haven, CT, USA), PerkinElmer Standard (Perkin Elmer, Waltham, MA, USA), or Merck Standard (Merck, Darmstadt, Germany). Pb, Cd, As, Sb, Ge, Co, Mn, Sn, and Hg were selected for analysis. The acetic acid used for elution was purchased from Merck (Darmstadt, Germany), and nitric acid was purchased from Eco Research (Republic of Korea). Deionized water was immediately purified using an EXL-7 water purification system (Vivagen, Republic of Korea).

### 2.3. Sample Preparation

The experiment was designed referring to the “Standards and Specifications for Utensils, Containers and Packages”, a law regulated by the Ministry of Food and Drug Safety in Korea [21]. To analyze the amount of heavy metals migrating from containers according to the elution time, 4% acetic acid heated to 70 °C was added to the container and covered with a watch glass; the temperature was maintained at 70 °C. The eluate was decanted at intervals of 5, 10, 20, 30, and 60 min and used as the test solution. Considering that food is eaten from these food containers, deionized water and 4% acetic acid solution were heated at 100 °C in a container, allowed to stand at 25 °C, poured at intervals of 1 h, 2 h, 1 d, and 2 d, and subsequently used as a test solution.

### 2.4. Instrumental Analysis

An ICP-MS spectrometer (Perkin Elmer, NexION 2000,Waltham, MA, USA) was used for quantitative analyses of the nine heavy metals and metalloid concentrations in the samples of deionized water and 4% acetic acid extracts. The device was equipped with a nickel cone, a cross-flow nebulizer, and a standard ICP quartz torch. Samples and standards were fed using a peristaltic pump. The spectrometer was optimized prior to analysis with a 1 μg/L solution (Be, Ce, Fe, In, Li, Mg, Pb, and U) in 1% HNO_3_ Setup Solution (Perkin Elmer). All target analytes, except As, were measured in the standard mode employing Ar. To determine As, dynamic reaction cell mode utilizing oxygen as the reaction gas was employed. The Hg content was measured using a direct mercury analyzer (Milestone, DMA-80, Italy). The instrumental settings used for the analysis are listed in Table 1.

### 2.5. Method Validation

Matrix analysis was used to validate sample preparation and analytical methods, linearity (expressed as R^2^), limit of detection (LOD), and limit of quantitation (LOQ). As the heavy metal content eluted through the food container was measured without intermediate treatment, the recovery rate was not determined. The analytical method was validated according to SANTE/11312/2021 [22] and ICH/2005/Q2/R1 [23]. The LOD and LOQ were calculated as the standard deviation of the slope and residual, respectively, using the regression line presented by the International Conference on Harmonization, and were evaluated according to the formula by repeating the procedure five times for each concentration using certified standard solutions.

### 2.6. Statistical Analyses

The eluted heavy metal contents (Pb, Sb, Cd, Ge, Co, Mn, Sn, As, and Hg) in deionized water and 4% acetic acid were used as the variables. The results were expressed as mean ± standard deviation. The relationships between the variables in the paper and plastic groups were calculated and compared using the independent samples *t*-test and repeated measures analysis of variance (ANOVA). The reported *p*-values were two-sided and considered statistically significant at 0.05 or less. The plastic group was further subdivided into PP, PS, and PET groups. The relationships between variables in the PP, PS, and PET groups were calculated and compared using the Kruskal–Wallis test and one-way ANOVA. Statistical significance was set at *p* < 0.05. All data analyses were performed using IBM SPSS Statistics for Windows (version 23.0; IBM Corp., Armonk, NY, USA).

## 3. Results and Discussion

### 3.1. Method Validation

Methods were validated for all metals and metalloids assessed in this study. Table 2 presents the linear correlation coefficients, LOD, and LOQ for the validation study. The recovery rate was excluded because the analysis was conducted immediately after dissolution from the food container. The linear correlation coefficient between the concentration and peak areas was in the range of 0.9919–0.9998, while LOD and LOQ were 0.0009–0.0176 and 0.0018–0.0582 μg/L, respectively. These results indicate that the experimental procedure was suitable for analyzing the metal content in food containers.

### 3.2. Migration of Metal and Metalloid Contents According to the Elution Time in 4% Acetic Acid at 70 °C

Figure 1 shows the results of the changes in the metal and metalloid content eluted from a container containing 4% acetic acid maintained at 70 °C. Analysis of the paper and plastic groups using repeated measures ANOVA indicated that Pb and Sb contents were substantially higher in the plastic group than in the paper group. In other words, the amount of Pb and Sb eluted over time was significantly higher in the plastic group compared to the paper group. An independent samples *t*-test was conducted to confirm the differences between the paper and plastic groups at each time point. In particular, Sb was consistently and strongly significantly higher in the plastic group than in the paper group at all times (*p* = 0.000). The content at each time point was found to be 0.014 ± 0.012 μg/L at 5 min, 0.014 ± 0.012 μg/L at 10 min, 0.016 ± 0.013 μg/L at 20 min, 0.021 ± 0.038 μg/L at 30 min, and 0.030 ± 0. 055 μg/L at 60 min in the paper group. In the plastic group, the content was 0.511 ± 0.591 μg/L at 5 min, 0.488 ± 0.589 μg/L at 10 min, 0.500 ± 0.623 μg/L at 20 min, 0.574 ± 0.723 μg/L at 30 min, and 0.669 ± 0.823 μg/L at 60 min. Additionally, the Co content was markedly higher in the plastic group at 5, 10, and 20 min; however, there was no difference between the groups over time. This suggests that the difference decreased with the increase in the amount of Co eluted from the paper containers.

Although a difference between the paper and plastic groups was not confirmed, many studies have confirmed that Sb is eluted into water from PET bottles [5,17,19]. This may have influenced the substantially higher Sb content in the plastic group than in the paper group in our study. Moreover, the Sb content significantly increased over time.

### 3.3. Comparison of Eluted Contents Using 4% Acetic Acid at 100 °C

Figure 2 shows the results of the changes over time in the amount of metals and metalloids transferred from the food container containing 4% acetic acid at 100 °C. The repeated measures ANOVA results revealed that the amount of Pb (*p* = 0.000), Sb (*p* = 0.000), Cd (*p* = 0.000), Co (*p* = 0.044), and Mn (*p* = 0.028) eluted from the plastic group was significantly higher than the paper group over time. An independent samples *t*-test was conducted to confirm the difference between the paper and plastic groups at each time point. Of the heavy metals analyzed, Sb (paper group: 1 h, 0.001 ± 0.003 μg/L; 2 h, 0.001 ± 0.002 μg/L; 1 d, 0.001 ± 0.002 μg/L; 2 d, 0.002 ± 0.005 *** μg/L; plastic group: 1 h, 1.108 ± 1.373 μg/L; 2 h, 1.194 ± 1.448 μg/L; 1 d, 1.175 ± 1.410 μg/L; 2 d, 1.216 ± 1.454 μg/L) and Cd (paper group: 1 h, 0.001 ± 0.002 μg/L; 2 h, 0.000 ± 0.001 μg/L; 1 d, 0.001 ± 0.002 μg/L; 2 d, 0.001 ± 0.002 μg/L; plastic group: 1 h, 0.007 ± 0.009 μg/L; 2 h, 0.005 ± 0.006 μg/L; 1 d, 0.005 ± 0.006 μg/L; 2 d, 0.005 ± 0.006 μg/L) were significantly higher in the plastic group at all times. These results are believed to be the result of little leaching of metals and metalloids into food from paper food containers, but greater leaching from plastic food containers.

Sb content was significantly higher in the plastic group, similar to the experimental results of elution by maintaining the containers in 4% acetic acid at 70 °C. This indicates that when 4% acetic acid is used as a food solvent, more Sb can be transferred into food from plastic food containers than paper food containers, regardless of elution time and storage method (warming; leaving).

Researchers in China simulated the potential release of Pb, Sb, Cd, Co, and Mn from food containers and confirmed that they were released at levels that were not hazardous to health [14]. These results are consistent with our results that indicated that Pb, Sb, Cd, Co, and Mn contents were increasingly leached from plastic food containers over time. Metals and metalloids are used as additives and catalysts in plastics to improve physical and chemical properties during processing [24,25]. Pb and Cd are mainly used as stabilizers [25], whereas Sb is typically used as a catalyst in plastic synthesis [17]. The use of metals and metalloids in plastic processing can explain their transfer from food containers to food.

According to the Korean Standards and Specifications for Utensils, Containers, and Packages, 4% acetic acid is used as the migration test solution for containers containing food with pH 5 or less [21]. Food containers are frequently used for delivering food and beverages. Although the pH range of the delivered foods is wide, many foods are acidic, including pasta sauce [26], kimchi [27], cheese [28], and vinegar [29]. Of the disposable food containers used for delivering beverages, take-out cups are frequently used and most beverages are acidic (average pH: 3.9) [30]. The pH of fruit and vegetable beverages (pH 3.1), carbonated beverages (pH 3.0), mixed beverages (pH 3.6) [30], yogurt drinks (pH 4.6), beer with 5% alcohol (pH 4.6) [29], and coffee (pH 5.26–5.32) varies depending on the extraction method [31]. Considering these food characteristics, the selected solvent was judged to be appropriate and sufficiently reflected the amount of transition of heavy metals to food.

### 3.4. Comparison of Eluted Contents Using Deionized Water at 100 °C

Figure 3 shows the results of elution using deionized water in the container at 100 °C. The repeated measures ANOVA results revealed a significant difference in Sb (*p* = 0.000), Co (*p* = 0.004), and Mn (*p* = 0.045) content. We used an independent samples *t*-test to confirm the difference between the paper and plastic groups at each time point. Statistically significant differences were identified in Sb and Co concentrations between the two groups. In particular, the elution of Sb was highly significant at 1 h, 2 h, 1 d, and 2 d (paper group: 1 h, 0.001 ± 0.002 μg/L; 2 h, 0.001 ± 0. 002 μg/L; 1 d, 0.001 ± 0.002 μg/L; 2 d, 0.001 ± 0.002 μg/L; plastic group: 1 h, 0.784 ± 0.959 μg/L; 2 h, 0.828 ± 1.008 μg/L; 1 d, 0.813 ± 0.997 μg/L; 2 d, 0.838 ± 1.026 μg/L).

The Sb content was found to be greatly significantly higher in the plastic group, similar to the results of the two types of dissolution experiments presented above. Based on the results in Figure 1, Figure 2 and Figure 3, the content of metals and metalloids that migrate into food was higher in the plastic group than in the paper group. In particular, in both of the food solvents 4% acetic acid and deionized water, more Sb eluted from plastic food containers regardless of the elution method (warming; leaving) and time. This result leads to a very interesting conclusion.

The Korean Standards and Specifications for Utensils, Containers, and Packages recommend food simulants for the migration test, which investigates the content of substances that leach out of food containers into food according to the characteristics of the contained food [21]. Oil–fat and fatty foods use n-heptane, alcoholic beverages use 20% alcohol, and all other foods use 4% acetic acid (food of not more than pH 5) or deionized water (food of more than pH 5) [21]. Most foods consumed for take-out in disposable containers fall into the other foods category. We conducted experiments using 4% acetic acid and deionized water to investigate the pH of all foods. The results lead to the conclusion that the migration of metals and metalloids from plastic food containers is high, regardless of the pH of the food, and that antimony in particular requires attention.

Therefore, paper packaging should be used rather than plastic packaging to prevent potential health hazards due to the migration of metals and metalloids from food containers. The results provide additional data for implementing policies that recommend reducing the use of plastic to prevent environmental pollution and using paper materials in disposable products.

### 3.5. Risk Assessment of Metals and Metalloids in Food Containers

According to the European Union’s Guidelines on Regulation (EU) No. 10/2011 of plastic materials and articles that come into contact with food, the overall migration of any substance into food must not exceed 60 mg/kg food or 10 mg/dm^2^ of the contact substance [32]. Our results showed concentrations below the overall migration limit.

A risk assessment was conducted using the estimated daily intake (EDI) and human exposure safety limits. EDI was calculated assuming that 2 L of water is consumed per day from a food container, based on the average dissolution of metals and metalloids. The human exposure safety limits for the risk assessment were as follows: We used the provisional maximum tolerable daily intake and provisional tolerable weekly intake (PTWI) values set by the Joint Food and Agriculture Organization of the United Nations/World Health Organization (WHO) Expert Committee on Food Additives. Cd was administered at 25 µg/kg body weight (bw)/month, Hg at 4 µg/kg bw/week, and Sn at 14 mg/kg bw/week. Pb was withdrawn in 2011 because establishing a scientifically acceptable Pb PTWI level for human health was not possible. In our study, the risk assessment was performed based on the past Pb PTWI value (25 μg/kg bw/week). Inorganic As followed the benchmark dose lower confidence limit of 0.5 as a standard of 3 μg/kg bw/d by WHO, and Mn was in the tolerable upper intake level of 11 mg/kg/d for adults, as given by the Korean Society of Nutrition. Co complied with the European Chemicals Agency’s General Population Hazard via oral route risk assessment, and long-term exposure produced no effect at 8.9 µg/kg bw/d. Sb was present at 6 μg/kg bw/d, which is a tolerable daily intake concentration established by the WHO for drinking water quality. Finally, Ge was excluded because no health risk standards have been established.

The hazard index is expressed as a percentage of EDI and the human exposure safety limit ratio. In our study, it ranged from 0.000% to 64.756% (Table 3). In particular, the hazard index obtained while maintaining the temperature at 70 °C was high compared to that at other temperatures. When keeping 4% acetic acid at 70 °C, lead showed higher values than other metals and metalloids (paper group: 44.744%; plastic group: 64.756%). In the case of antimony, we found a low value of 1.003% in paper and a relatively high value of 22.296% in plastic. In the case of 100 °C water or 4% acetic acid, the highest hazard index for Sb was found in the plastic group (deionized water: 27.933%/4%; acetic acid: 20.267%).

A hazard index over 100% indicates that exposure to metals can induce obvious toxic effects. All values in our study were within 100%,and were judged not to be hazardous to human health. Nevertheless, substances such as Pb and Sb, which have high hazard index values, require caution when using food containers.

Our study took into account the amount of time people actually spend putting food in food containers. In addition, an experiment was conducted using 4% acetic acid and deionized water to reflect the influence of the pH of the food contained in the containers. However, various ingredients in different food items may affect the dissolution of metals and metalloids; additionally, the risk exposure to humans may differ.

The amounts of metals and metalloids in food containers used to package food for delivery and take-out were distributed at a safe level. However, to minimize exposure to trace amounts of some metals and heavy metals, it is recommended that food be consumed immediately after being placed in a food container. Additionally, when eating delivered or packaged food, transferring it to a container rather than storing it in a packaging container for a long period is suggested to reduce exposure to metals and metalloids.

## 4. Conclusions

The amount of metals and metalloids migrating from plastic and paper food containers to food was compared. In total, 120 paper and plastic food containers were filled with 4% acetic acid and deionized water, and the migration of metals and metalloids eluted over time was confirmed. The disposable paper and plastic food containers distributed in South Korea used in our study were confirmed to be not harmful to humans by a risk assessment (the range of the hazard index was 0.000–64.756%). However, the eluted metals and metalloids (lead, antimony, cadmium, cobalt, etc.) in plastic containers were significantly higher. In particular, the migration of Sb was higher in plastic food containers than in paper containers, regardless of the solvent or time. Although food containers are widely used worldwide, research on metals and metalloids that can be eluted from food containers is limited, and further studies are needed. In this respect, our study provides baseline results for the migration of metals and metalloids from food containers to food and suggests an experimental direction for the migration of food substances that can potentially occur in food containers. The results provide useful data for appropriate and safe use of food containers.

## Figures and Tables

**Figure 1 ijerph-21-00139-f001:**
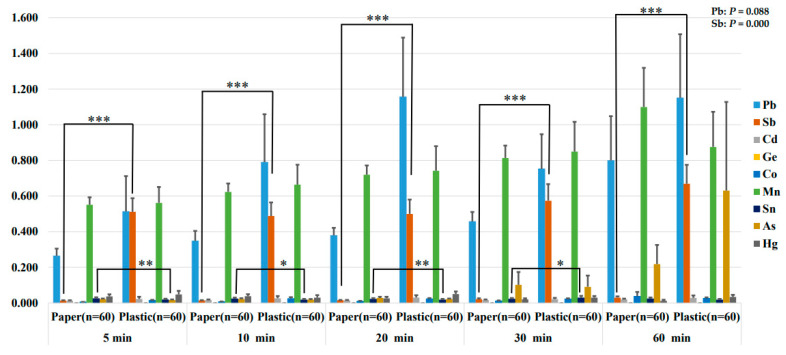
Metal and metalloid elution content of paper and plastic food containers over time in 4% acetic acid maintained at 70 °C (μg/L). All values are mean, and air bars are standard deviation. Leaching of metals and metalloids from paper and plastic groups over time was evaluated by repeated measures ANOVA analysis (Pb: *p* = 0.088; Sb: *p* = 0.000). Differences at each time point were determined using independent samples *t*-test (*p* < 0.05 *, *p* < 0.01 **, and *p* < 0.001 ***).

**Figure 2 ijerph-21-00139-f002:**
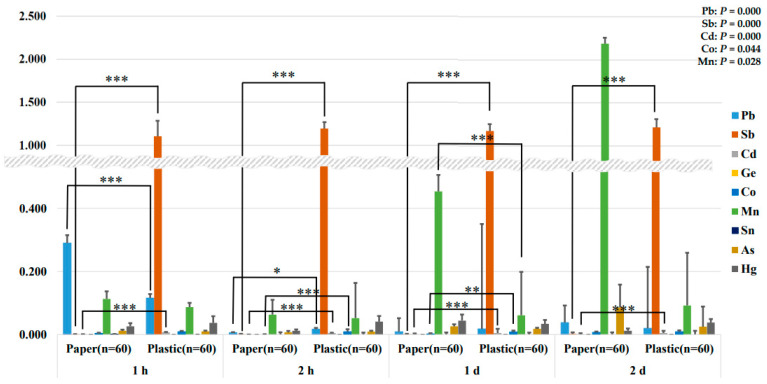
Analysis of metal and metalloid elution content of paper and plastic food containers over time after being left in 4% acetic acid at 100 ° C (μg/L). All values are mean, and air bars are standard deviation. Leaching of metals and metalloids from paper and plastic groups over time was evaluated by repeated measures ANOVA analysis (Pb: *p* = 0.000, Sb: *p* = 0.000, Cd: *p* = 0.000, Co: *p* = 0.044, and Mn: *p* = 0.028). Differences at each time point were determined using independent samples *t*-test (*p* < 0.05 *, *p* < 0.01 **, and *p* < 0.001 ***).

**Figure 3 ijerph-21-00139-f003:**
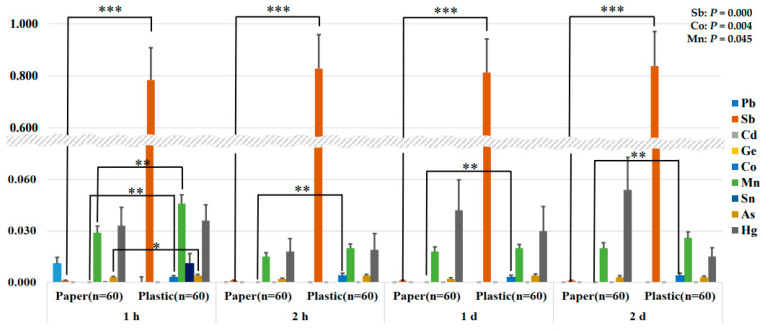
Analysis of metal and metalloid elution content of paper and plastic food containers over time after being left in deionized water at 100 °C (μg/L). All values are mean, and air bars are standard deviation. Leaching of metals and metalloids from paper and plastic groups over time was evaluated by repeated measures ANOVA analysis (Sb: *p* = 0.000, Co: *p* = 0.004, and Mn: *p* = 0.045). Differences at each time point were determined using independent samples *t*-test (*p* < 0.05 *, *p* < 0.01 **, and *p* < 0.001 ***).

**Table 1 ijerph-21-00139-t001:** Instrument operating conditions for analyzing determination by ICP-MS and direct mercury analyzer.

ICP-MS Spectrometer		
Parameter	Operation Conditions	
	Standard	Oxygen DRC
Nebulizer Gas Flow	0.99 L/min	1 L/min
Auxiliary Gas Flow	1.2 L/min	1.2 L/min
Plasma Gas Flow	15 L/min	15 L/min
RF power	1600 W	1600 W
Residence Time	50 µs	50 µs
Analysis Time	188 s	188 s
**Direct Mercury Analyzer**		
**Parameter**	**Operation Conditions**	
Drying temperature	200 °C	
Drying time	120 s	
Decomposition temperature	650 °C	
Decomposition time	210 s	
Amalgamator heating temperature	120 °C	
Amalgame time	12 s	
Recording time	30 s	

**Table 2 ijerph-21-00139-t002:** Validation parameters (linearity, LOD, and LOQ) in our study.

	Correlation Coefficient (R^2^)	LOD (μg/L) ^a^	LOQ (μg/L) ^b^
Pb	0.9945	0.0009	0.0027
Sb	0.9988	0.0021	0.0063
Cd	0.9998	0.0030	0.0090
Ge	0.9963	0.0060	0.0018
Co	0.9943	0.0024	0.0072
Mn	0.9965	0.0021	0.0063
Sn	0.9961	0.0006	0.0018
As	0.9919	0.0150	0.0315
Hg	0.9977	0.0176	0.0582

LOD: limit of detection; LOQ: limit of quantification. ^a^ Limit of detection (LOD) was calculated as 3.3× standard deviation of the response/slope of calibration curve. ^b^ Limit of quantification (LOQ) was determined as 10× standard deviation of the response/slope of calibration curve.

**Table 3 ijerph-21-00139-t003:** Risk assessment of metals and metalloids leached from food containers.

Maintained in 4% Acetic Acid at 70 °C (5, 10, 20, 30, and 60 min)						
	Average Concentration (μg/L)		EDI ^a^ (μg/person/day)		Hazard Index ^b^ (%)	
	Paper (*n* = 60)	Plastic (*n* = 60)	Paper (*n* = 60)	Plastic (*n* = 60)	Paper (*n* = 60)	Plastic (*n* = 60)
Pb	0.264~0.799	0.513~1.156	0.528~1.598	1.026~2.313	14.784~44.744	28.734~64.756
Sb	0.014~0.030	0.488~0.669	0.027~0.060	0.976~1.148	0.453~1.003	16.265~22.296
Cd	0.012~0.020	0.031~0.021	0.024~0.040	0.043~0.061	2.903~4.787	5.173~7.347
Co	0.005~0.037	0.015~0.026	0.011~0.075	0.030~0.052	0.120~0.838	0.332~0.588
Mn	0.551~1.100	0.562~0.875	1.102~2.199	1.124~1.751	0.010~0.020	0.010~0.016
Sn	0.021~0.023	0.017~0.028	0.048~0.042	0.033~0.056	0.002~0.002	0.002~0.003
As	0.021~0.218	0.016~0.631	0.043~0.436	0.033~1.262	1.424~14.540	1.091~42.051
Hg	0.012~0.037	0.030~0.050	0.024~0.074	0.060~0.100	4.200~13.300	10.500~17.500
**Maintained in 4% Acetic Acid at 100 °C (1 h, 2 h, 1 d, and 2 d)**						
	**Average Concentration (μg/L)**		**EDI (μg/person/day)**		**Hazard Index (%)**	
	**Paper (*n* = 60)**	**Plastic (*n* = 60)**	**Paper (*n* = 60)**	**Plastic (*n* = 60)**	**Paper (*n* = 60)**	**Plastic (*n* = 60)**
Pb	0.290~0.006	0.017~0.116	0.580~0.012	0.034~0.232	0.168~8.120	0.476~3.248
Sb	0.001~0.002	1.108~1.194	0.002~0.004	2.216~2.388	0.017~0.033	18.467~20.267
Cd	0.000~0.001	0.005~0.007	0.000~0.002	0.010~0.014	0.000~0.120	0.600~0.840
Co	0.000~0.007	0.008~0.009	0.000~0.014	0.016~0.018	0.000~0.079	0.090~0.101
Mn	0.113~2.188	0.052~0.092	0.226~4.376	0.104~0.184	0.573~19.891	0.473~0.836
Sn	0.000~0.001	0.000~0.000	0.000~0.002	0.000~0.000	0.000~0.050	0.000~0.000
As	0.007~0.086	0.009~0.025	0.014~0.172	0.018~0.050	0.233~2.867	0.300~0.833
Hg	0.012~0.026	0.037~0.041	0.024~0.052	0.074~0.082	2.100~7.700	5.950~7.175
**Maintained in Deionized Water at 100 °C (1 h, 2 h, 1 d, and 2 d)**						
	**Average Concentration (μg/L)**		**EDI (μg/person/day)**		**Hazard Index (%)**	
	**Paper (*n* = 60)**	**Plastic (*n* = 60)**	**Paper (*n* = 60)**	**Plastic (*n* = 60)**	**Paper (*n* = 60)**	**Plastic (*n* = 60)**
Pb	0.000~0.011	0.000~0.000	0.000~0.022	0.000~0.000	0.000~0.616	0.000~0.000
Sb	0.001~0.001	0.784~0.838	0.002~0.002	1.568~1.676	0.033~0.033	26.133~27.933
Cd	0.000~0.000	0.000~0.000	0.000~0.000	0.000~0.000	0.000~0.000	0.000~0.000
Co	0.000~0.000	0.003~0.004	0.000~0.000	0.006~0.008	0.000~0.000	0.067~0.090
Mn	0.015~0.029	0.020~0.046	0.030~0.058	0.040~0.092	0.000~0.001	0.000~0.001
Sn	0.000~0.000	0.000~0.011	0.000~0.000	0.000~0.022	0.000~0.000	0.000~0.001
As	0.002~0.003	0.003~0.004	0.004~0.006	0.006~0.008	0.133~0.200	0.200~0.267
Hg	0.018~0.054	0.015~0.036	0.036~0.108	0.030~0.072	6.300~18.900	5.250~12.600

EDI: estimated daily intake. ^a^ Average concentration (μg/L) × 2L (daily water intake). ^b^ Hazard index (%) = (EDI/safety limits for risk assessment) × 100.

## Data Availability

The data presented in this study are available on request from the corresponding author. The data are not publicly available due to further internal research.

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
