# Peer review of "Potential Migration and Health Risks of Heavy Metals and Metalloids in Take-Out Food Containers in South Korea"

_ijerph, 2024, doi:10.3390/ijerph21020139_

Round 1

Reviewer 1 Report

Comments and Suggestions for Authors

Abstract: Mention key finding results in the abstract section. For example in the line ¨We found that the eluted amount of 20 metals and metalloids in food was substantially higher in the plastic group.¨, provide the results of metal concentrations in both paper and plastic types of food containers.

Introduction must strongly base the details on studies conducted on migration of metals in food containers so far in literature.

The need for current study must be justified in depth.

In results and discussion section, wherever adequate provide results (concentrations of metals).

A separate section before conclusion should be provided for in depth discussions of metal concentration variation in food containers based on temperature and intervals of time.

In conclusion risk assessment results and conclusions must be described.  

All the results are presented in table format. Th authors should consider presenting graphical illustrations for better representation of the work.

Check for typo errors throughout the manuscript.

Author Response

Thank you for reviewing our paper. Also, your advice has helped us to improve our paper for the better.

I have marked the modified parts of the manuscript in red.

1. Abstract: Mention key finding results in the abstract section. For example in the line ¨We found that the eluted amount of 20 metals and metalloids in food was substantially higher in the plastic group.¨, provide the results of metal concentrations in both paper and plastic types of food containers.

Thanks for the advice. I have made an effort to clarify the results as mentioned. That is, I have simplified the methods in the abstract and increased the numbers and discussion of the results.

2. Introduction must strongly base the details on studies conducted on migration of metals in food containers so far in literature.

Thank you for your advice.

I have taken your suggestion and added a literature review on food containers to the introduction of the manuscript. This helped to clarify the need for our study.

3. The need for current study must be justified in depth.

 Currently, people around the world are increasingly using disposable food containers. This means that it is necessary to investigate the substances that can be migrated out of food containers depending on how long they are stored and eaten. However, there are insufficient studies in this field, so we aimed to compare the substances that can migrate from paper and plastic, which are commonly used as materials for food containers. I have revised the introduction to clarify in more depth the need for the current study.

4. In results and discussion section, wherever adequate provide results (concentrations of metals).

As advised, we have modified the results and discussion to provide information such as exact numbers without looking at the table.

5. A separate section before conclusion should be provided for in depth discussions of metal concentration variation in food containers based on temperature and intervals of time.

We have described in each section the difference in elution between paper and plastic group at each time. We have also done a statistical analysis of the time x group.

Are you referring to the results of looking at temperature alone, regardless of group? We excluded this because we were not comparing between groups. However, we did have the statistical results for that part.

If this needs to be added, it shouldn't be difficult for us to do so.

Please give me your advice on this.

6. In conclusion risk assessment results and conclusions must be described.  

   I revised the content of the conclusion, adding the results of the risk assessment to the conclusion.

7. All the results are presented in table format. The authors should consider presenting graphical illustrations for better representation of the work.

I changed Tables 3,4,5 to figures 1,2,3 as suggested. I hope this helps others understand the data.

8. Check for typo errors throughout the manuscript.

I check for typo errors throughout the manuscript.

Reviewer 2 Report

Comments and Suggestions for Authors

Overall, this manuscript is well written and reflect the title. However, several comments suggested can be addressed to enhance the quality of the journal.

Line 82 – Why specifically choose those two types of samples? How about polystyrene?

Line 86 – What are the Certified reference standards using in this study?

Line 139 – Provide data from the Certified reference standards.

Line 177 – Are they using the same type of plastic?

Line 220 – Author calculate the PTWI value for the risk assessment. Any citation or previous study using the same approach? Normally, the PTWI is calculate when we consume the sample.

Some data in the table suggested to presented in graph form, in order to easy to understand.

Author Response

Thank you for taking the time to review our paper. I have prepared the following answers to your queries.

Line 82 – Why specifically choose those two types of samples? How about polystyrene?

We aimed to identify and compare the leaching of metals and metalloids from disposable synthetic resins and paper. All over the world, plastic and paper are used as disposable food containers. In particular, plastics have a long degradation period and can be discharged into rivers and oceans, causing not only environmental pollution but also threatening human health. Based on this, many countries are recommending reducing the use of plastics. We analyzed the elution of metals and metalloids from synthetic resin and paper, and emphasized the high content of metals and metalloids in plastics to recommend the use of paper. Based on these results, we will conduct future research on the types of plastics, PP, PET, PS, etc. and study which material elutes more substances.

Line 86 – What are the Certified reference standards using in this study?

Certified Reference Standards are a high-level standard that provides a robust statistical evaluation of the certified value for the analyte of interest. Complete metrological traceability is included with this product and a practical representation of uncertainty provided. In other words, the use of these Certified Reference Standards (CRMs) means that the data we are experimenting with has a proven standardization, and we can trust our results.

Line 139 – Provide data from the Certified reference standards.

Method validation is the process of proving that an analytical method is appropriate for its intended purpose, meaning that we can use certified standards to prove that the values we analyze are accurate. We performed our own method validation using these standards and obtained the values in Table 2.

Line 177 – Are they using the same type of plastic?

The study was conducted in China and used take-out food containers made of plastic and paper to investigate leaching. Similar to our results, high values of cadmium, antimony, manganese, and cobalt were observed in the plastic material sample.

Line 220 – Author calculate the PTWI value for the risk assessment. Any citation or previous study using the same approach? Normally, the PTWI is calculate when we consume the sample.

The Provisional Tolerance Weekly Intake (PTWI) is a value applied to a food contaminant to indicate the amount that can be consumed weekly over a lifetime without significant health effects. This is a common threshold used in studies of metals that can be ingested through food. Currently, various papers have used it to evaluate health risks, including the following

  1. Lee, HyoMin, et al. "Health risk assessment of lead in the Republic of Korea." Human and Ecological Risk Assessment9.7 (2003): 1801-1812.

  2. Alva, Camila Valente, et al. "Concentrations and health risk assessment of total mercury in canned tuna marketed in Southest Brazil." Journal of Food Composition and Analysis88 (2020): 103357

  3. Pirsaheb, Meghdad, et al. "Essential and toxic heavy metals in cereals and agricultural products marketed in Kermanshah, Iran, and human health risk assessment." Food Additives & Contaminants: Part B9.1 (2016): 15-20.

Similar to these papers, my study used some PWTI criteria to determine whether metals and metalloids eluted from single-use containers are a health hazard.

Some data in the table suggested to presented in graph form, in order to easy to understand.

I changed Tables 3,4,5 to figures 1,2,3 as suggested. I hope this helps others understand the data.

Reviewer 3 Report

Comments and Suggestions for Authors

This manuscript presented solid data, but improvement has to be made to warrant its acceptance to this journal. I have the following concerns:

1.     Specific experimental procedures do not need to be given in the abstract, and more conclusions of this study should be presented.

2.     L 39, (Raheem, 2013). Please unify the format of references.

3.     In this work, the authors collected 60 paper samples and 60 plastic samples, and found that the eluted amount of metals and metalloids in food was substantially higher in the plastic group. As the authors mentioned in the Introduction section, “Of the plastic materials, polypropylene (PP), polystyrene (PS), polyethylene terephthalate (PET), and polyethylene are commonly used materials in food containers”.

Therefore, as readers and consumers, people may care more about that which type of plastic makes disposable food containers riskier, or which type of containers, such as food or drinks, are riskier. We would suggest adding a section in the discussion to explore these concerns.

4.     In table 2, show the linearity range.

5.     Tables 3-5, we would suggest adding figures for better presentation of these data.

Comments on the Quality of English Language

Moderate editing of English language required

Author Response

Thank you for reviewing our paper. Also, your advice has helped us to improve our paper for the better.

1. Specific experimental procedures do not need to be given in the abstract, and more con clusions of this study should be presented.

Thanks for the advice. I have made an effort to clarify the results as mentioned. That is, I have simplified the methods in the abstract and increased the numbers and discussion of the results.

I've colored the changes in red in the word file, and the changes look like this, with the text darker.

The consumption of take-out food has increased worldwide; consequently, people are increasingly being exposed to chemicals from food containers. However, research on the migration of metals from containers to food is limited, and therefore, information required to determine the health risks is lacking. Herein, the amount of transfer nine metals and metalloids (Pb, Sb, Cd, Ge, Co, Mn, Sn, As, and Hg) from food containers to food in South Korea was assessed from take-out food containers classified as paper and plastic container groups. The sample containers were eluted over time by either warming with 70 °C 4% acetic acid or cooling with 100 °C 4% acetic ac-id/deionized water at 25°C. It was analyzed using an inductively coupled plasma mass spectrometer and a direct mercury analyzer. The reliability of the quantitative results was verified by calculating the linearity, limit of detection, and limit of quantification. We found that the amount of metals and metalloids (Pb, Sb, Cd, and Co) eluting over time was highly significant in plastic group. Regardless of the food simulant and elution time, the amount of Sb transferred from the food containers to food was substantially higher in the plastic (average concentration: 0.488-1.194㎍/L) than in the paper group (average concentration: 0.001-0.03㎍/L). Fortunately, all food con-tainers were distributed at levels safe for human health (harzard index: 0.000-64.756 %). However, caution is needed when warm food is added to food containers. Overall, our results provide base-line data for the management and use of take-out containers.

2. L 39, (Raheem, 2013). Please unify the format of references.

    I unify the format of references.

3. In this work, the authors collected 60 paper samples and 60 plastic samples, and found that the eluted amount of metals and metalloids in food was substantially higher in the plastic group. As the authors mentioned in the Introduction section, “Of the plastic materials, polypropylene (PP), polystyrene (PS), polyethylene terephthalate (PET), and polyethylene are commonly used materials in food containers”. Therefore, as readers and consumers, people may care more about that which type of plastic makes disposable food containers riskier, or which type of containers, such as food or drinks, are riskier. We would suggest adding a section in the discussion to explore these concerns.

We aimed to identify and compare the leaching of metals and metalloids from disposable synthetic resins and paper. All over the world, plastic and paper are used as disposable food containers. In particular, plastics have a long degradation period and can be discharged into rivers and oceans, causing not only environmental pollution but also threatening human health. Based on this, many countries are recommending reducing the use of plastics. We analyzed the elution of metals and metalloids from synthetic resin and paper, and emphasized the high content of metals and metalloids in plastics to recommend the use of paper. Based on these results, we will conduct future research on the types of plastics, PP, PET, PS, etc. and study which material elutes more substances. We have excluded this part of the study in order to add another study using the plastic containers used in the current study. However, if it is considered essential, we will look into it further internally.

4. In table 2, show the linearity range.

 I know that the Correlation Coefficient (R2) in Table 2 is the linearity value. Therefore, we wrote "The linear correlation coefficient between the concentration and peak areas was in the range of 0.9919-0.9998" in the manuscript. Also, all values are above 0.99 and close to 1. Therefore, we can judge that there is linearity. If you mean linearity range in a different sense, let me know and I'll do it.

5. Tables 3-5, we would suggest adding figures for better presentation of these data.

I changed Tables 3,4,5 to figures 1,2,3 as suggested.

Round 2

Reviewer 1 Report

Comments and Suggestions for Authors

there are no additional comments to add, the manuscript has been substantially improved and it is recommended for its acceptance and publication.